# CaNOCS: Category-Level 3D Correspondence from a Single Image

## Abstract

Recent progress in 6D object pose estimation has been driven by representations that map image pixels to normalized object coordinate spaces (NOCS). However, NOCS representations are fundamentally tailored to pose estimation, but are insufficient for detailed object understanding, since the same point in NOCS space may correspond to different semantic parts across object instances. We argue that the next frontier in object understanding is **category-level 3D correspondence**: predicting, from a single image, the canonical 3D location of each pixel in a way that is semantically aligned across all instances of a category. Such correspondences go beyond pose—they enable reasoning about function and interaction. To enable research in this direction, we introduce **HouseCorr3D**, the first dataset with dense semantic 2D–3D correspondences across 50 household object categories, including annotated CAD models, hundreds of real images per class, and amodal correspondences for occluded regions. We further propose **CaNOCS**, a framework for learning category-level **morphable shape priors** to enable 3D correspondence estimation that is semantically aligned across category instances. In extensive experiments, CaNOCS achieves substantially better category-level 3D correspondence than baselines based on NOCS or DINOv2. We believe that CaNOCS and HouseCorr3D establish a new paradigm to move beyond the 6D pose toward **fine-grained, correspondence-level object understanding** with broad applications in robotics and AR/VR.

## 1 Introduction

Understanding 3D objects from images is a long-standing challenge in computer vision, with applications in robotics, augmented reality (AR), and virtual reality (VR). So far, a main focus in this area has been the development of methods for 6D object pose estimation, which predict the 3D translation and rotation of objects relative to the camera (Xiang et al., 2018; Tekin et al., 2018; He et al., 2020). In this context, an important challenge is to generalize across unseen instances of a given object category, and the *Normalized Object Coordinate Space (NOCS)* (Wang et al., 2019) representation is the de facto standard approach to address this problem. In NOCS, each pixel of an object is mapped to a normalized 3D coordinate system that is shared across all instances of a category, hence enabling *category-level* 6D pose estimation methods and benchmarks (Liu et al., 2020; Li et al., 2020; Chen et al., 2020; Rodrigues et al., 2022; Lin et al., 2025; 2023).

However, NOCS representations are fundamentally tailored to pose estimation rather than detailed object understanding. They provide normalized coordinates per pixel, but do not guarantee *category-consistent 2D–3D correspondences*. As a result, each instance can drift in its own canonicalization, so that the same NOCS coordinate may correspond to different semantic parts across objects of the same category as illustrated in Figure 4. This ambiguity is especially problematic under large intra-class variation, symmetry, and occlusion. A consequence of this limitation is that NOCS-style approaches are effective for recovering coarse pose, but insufficient for downstream tasks that require fine-grained *semantic correspondence across instances*—such as transferring grasp strategies in robotics or aligning CAD assets in AR/VR.

We argue that the next frontier in object understanding is *category-level 3D correspondence*: predicting for each pixel of an object instance, a canonical 3D location in a way that is semantically aligned across all instances of a category (Figure 1). Such category-level 3D correspondences ex-

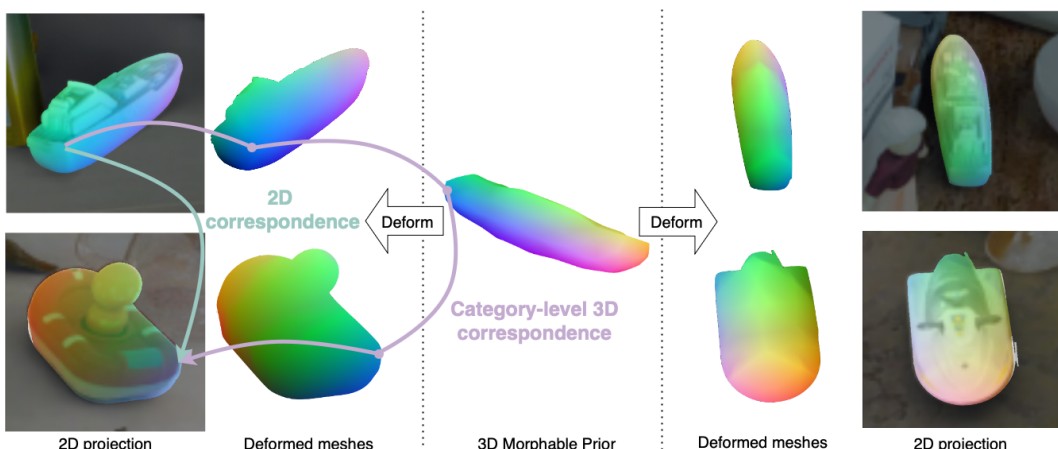

2D correspondence

Deform

Category-level 3D correspondence

Deform

2D projection   Deformed meshes   3D Morphable Prior   Deformed meshes   2D projection

Figure 1: We introduce the task of category-level 3D correspondence, which we make measurable through our benchmark HouseCorr3D providing cross-instance 3D semantic annotations. To address this task, we propose CaNOCS, a method that anchors NOCS-style (Wang et al., 2019) representations to a canonical 3D deformable prior, thereby enforcing semantic consistency across instances.

tend beyond 6D pose. They provide a unified canonical space that links geometry and semantics, enabling reasoning about object function, interaction, and affordances.

However, existing benchmarks are not designed to evaluate this capability. Current datasets such as NOCS-Real275 (Wang et al., 2019), Wild6D (Rodrigues et al., 2022), OmniNOCS (Lin et al., 2025), and Omni6DPose (Lin et al., 2023) provide pose annotations, segmentation, and depth, but they *do not annotate category-level 3D correspondences*. This makes it impossible to measure whether methods actually learn correspondences that are consistent across instances—a key limitation of NOCS-style methods. To address this gap, we introduce **HouseCorr3D**, the first dataset with category-level 2D–3D correspondences across 50 everyday object categories. HouseCorr3D includes annotated CAD models, hundreds of real images per class, and *amodal correspondences* that annotate occluded object parts (see Figure 3). These annotations enable, for the first time, quantitative evaluation of category-level 3D correspondence understanding.

Building on our benchmark, we propose **CaNOCS (Canonicalized NOCS)**, a framework that extends NOCS-style 6D pose estimation with *morphable category priors*. In particular, CaNOCS learns deformable canonical 3D models that provide semantically consistent mappings across instances of an object category. Figure 1 illustrates our problem setting, and our proposed category-level canonicalization that enforces semantic consistency across instances. In summary, our contributions are as follows:

(i) We identify **category-level 3D correspondence** as the next frontier in object understanding, moving beyond pose to canonicalized, semantically aligned 3D coordinates across instances.

(ii) We release **HouseCorr3D**, the first benchmark for category-level 2D–3D correspondence across everyday objects, covering 50 categories with annotated CAD models, hundreds of real images per class, and amodal correspondence labels.

(iii) We propose **CaNOCS**, which extends NOCS-style representations with morphable 3D priors to enforce a consistent canonicalization across object instances.

(iv) We demonstrate that CaNOCS substantially outperforms NOCS- and DINOv2-based baselines on HouseCorr3D, establishing a new paradigm for *fine-grained, correspondence-level object understanding* with applications in robotics and AR/VR.

## 2 RELATED WORK

**6D Pose Estimation and NOCS.** Benchmarks for 6D pose estimation, such as LINEMOD (Hinterstoisser et al., 2012), Occlusion LINEMOD (Brachmann et al., 2014), YCB-Video (Xiang et al., 2018), T-LESS (Hodan et al., 2017), HomebrewedDB (Kaskman et al., 2019), and the BOP challenge (Hodan et al., 2020), focus on rigid objects under controlled capture conditions with precise pose annotations. Datasets like Pascal3D+ (Xiang et al., 2014), ImageNet3D (Ma et al., 2024), and Omni3D (Brazil et al., 2023) extend these efforts to natural images but remain limited to sparse labels without category-level 3D correspondence. NOCS datasets (Wang et al., 2019; Wen et al., 2023) introduced normalized coordinate spaces that generalize across instances, though they are designed primarily for pose estimation rather than correspondence.

**Category-Level Correspondence.** Category-level 3D correspondence aims to map image pixels to canonical surfaces shared across instances. It has been approached through two main strategies: deformation-based methods that map instances to a template mesh (Groueix et al., 2018; Wang et al., 2018), and template-free approaches that learn canonical coordinate systems without reliance on a single exemplar (Novotny et al., 2019). More recent work leverages large-scale features and foundation models for semantic alignment across categories (Neverova et al., 2020; Shtedritski et al., 2024). Dense correspondences have also been explored in narrower domains such as humans (Güler et al., 2018) and quadrupeds (Xu et al., 2023), but these methods lack category diversity and amodal supervision for everyday objects.

**Morphable Models and Shape Priors.** Morphable models capture intra-class shape variability by deforming canonical templates. Classic work addressed faces and human bodies (*e.g.,*, 3D Morphable Models (Blanz & Vetter, 1999), SMPL (Loper et al., 2015)). More recent approaches (Neverova et al., 2020; Shtedritski et al., 2024; Sommer et al., 2025; Kim et al., 2024) extend these ideas to more object classes or generative deformation guided by diffusion priors. These methods show that morphable models are powerful for representing shape variation, yet generalizing these methods to general scenes with dozens of everyday objects remains an open challenge.

**Datasets for Category-Level Understanding.** Large-scale 3D shape collections such as ModelNet (Wu et al., 2015) and ShapeNet (Chang et al., 2015) provide broad category coverage with clean CAD meshes, while part-annotated datasets like ShapeNetPart (Yi et al., 2016) and PartNet (Mo et al., 2019) add hierarchical or per-point part labels that enable structured reasoning over geometry. These resources have been instrumental for learning accurate shapes, part segmentation, shape completion, and part-aware priors, and they support tasks such as part-level manipulation or assembly. However, these datasets lack consistent correspondences between distinct instances at the point or surface level, which limits their usefulness for tasks that require cross-instance alignment. More recently, Omni6DPose (Lin et al., 2023) integrated ShapeNet-like CAD models into realistic synthetic scenes, enabling pose estimation approaches to learn from photorealistic renderings rather than isolated meshes. Similarly, CO3D (Reizenstein et al., 2021), Pix3D (Sun et al., 2018), and NOCS (Wang et al., 2019) provide real or synthetic multi-view images with pose labels, but they still do not supply semantic, amodal, point-level correspondences across diverse instances.

**Correspondence Matching and Self-Supervised Features.** 2D correspondence has advanced from local descriptors and dense flows (e.g., SIFT (Lowe, 2004), DAISY (Tola et al., 2010), SIFT Flow (Liu et al., 2011), DeepFlow (Weinzaepfel et al., 2013)) to transformer-based self-supervised features (Caron et al., 2021; Zhou et al., 2022; Oquab et al., 2023), which exhibit emergent alignment and achieve strong results on SPair-71K, PF-PASCAL, and TSS (Zhang et al., 2023; Li et al., 2023). Dedicated matchers such as LoFTR, COTR, DiffMatch (Sun et al., 2021; Jiang et al., 2021; Nam et al., 2024), and spherical-map approaches (Mariotti et al., 2024) further improve dense matching, with benchmarks like PF-PASCAL, PF-WILLOW, SPair-71K (Min & Cho, 2019; Ham et al., 2016) and MISC210K (Sun et al., 2023) broadening coverage. These methods achieve strong pairwise correspondences but do not yield metrically grounded canonical coordinates or enforce semantic consistency in 3D. Complementary efforts in 3D, such as DenseMatcher (Zhu et al., 2024), extend matching to the mesh domain by projecting multiview features onto 3D geometry and solving for correspondences via functional maps, but they still fall short of providing image-to-canonical or amodal correspondences across diverse categories.

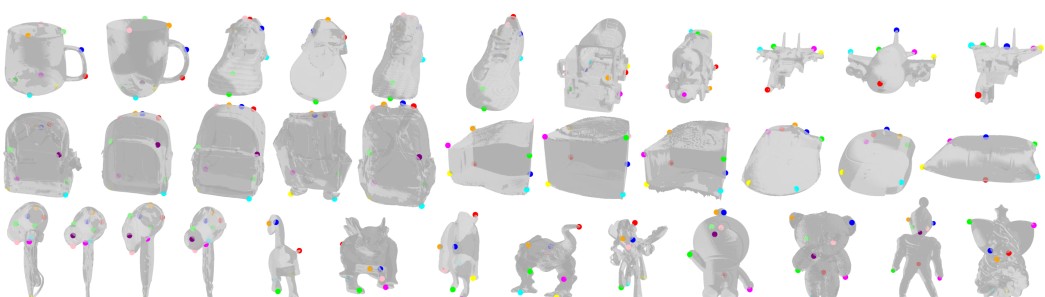

Figure 2: **Dataset annotation overview.** We highlight annotated keypoints for a set of instances.

## 3 THE HOUSECORR3D CORRESPONDENCE BENCHMARK

We introduce **HouseCorr3D**, the first benchmark for *category-level 3D correspondences*. Existing datasets focus either on 6D pose (Wang et al., 2019; Lin et al., 2025; 2023) or 2D semantic correspondence evaluation Min & Cho (2019) and provide no means to measure whether predicted correspondences are semantically consistent across object instances in 3D. HouseCorr3D fills this gap by enabling, for the first time, a quantitative assessment of category-level 3D correspondence.

**Data selection and annotation.** We build HouseCorr3D based on Omni6DPose (Lin et al., 2023), selecting 50 everyday categories such as mugs, bottles, and remotes. For each, a few representative CAD meshes are annotated with 3D semantic keypoints shared across all instances, enabling the evaluation of *category-level* 3D correspondence (see Figure 2). Using ground-truth poses, these annotations are automatically projected into all image views, yielding consistent 2D correspondences across thousands of appearances with minimal manual effort. This mesh-centric strategy enforces semantic consistency across objects and poses, provides *amodal* labels for occluded regions, and scales a compact set of 3D annotations into a large benchmark of 2D and 3D correspondences.

**Symmetry handling.** In HouseCorr3D, we explicitly account for object symmetries in correspondence evaluation, focusing on *rotational* and *reflective* cases.
*Rotational symmetry* is defined as invariance under rotations about a fixed axis. For an object with $N$-fold rotational symmetry (*e.g.,* unchanged by $2\pi/N$ rotations), the valid transformations form the cyclic group $C_N = \{R_{\mathbf{a}}(2\pi k/N)\}_{k=0}^{N-1}$, with $N \to \infty$ for continuous symmetry. A ground-truth keypoint $\boldsymbol{x}$ then has multiple equivalent positions—its *orbit* generated by rotations around the axis. We treat all points on this orbit as valid correspondences (see Figure 3b), resolving ambiguity when a prediction could match any rotated instance of $X$. Given a predicted point $\hat{\boldsymbol{x}}$ and rotation $R_{\mathbf{a}}(\theta)$ about axis $\mathbf{a}$, the correspondence error is $e_{\text{rot-sym}}(\boldsymbol{x}, \hat{\boldsymbol{x}}) = \min_\theta \|R_{\mathbf{a}}(\theta)\, \boldsymbol{x} - \hat{\boldsymbol{x}}\|$, with $\theta \in [0, 2\pi)$ (or $\{2\pi k/N\}$). Geometrically, this equals the distance from $\hat{\boldsymbol{x}}$ to the circular locus of $\boldsymbol{x}$ around the axis.
*Reflective symmetry* is invariance under a mirror reflection across a plane. If an object is symmetric about plane $\Pi$, each keypoint $\boldsymbol{x}$ has a mirrored counterpart $\boldsymbol{x}'$. We treat $\boldsymbol{x}$ and $\boldsymbol{x}'$ as equivalent, so a prediction may match either without penalty Figure 3a. The error is $e_{\text{refl-sym}}(\boldsymbol{x}, \hat{\boldsymbol{x}}) = \min\{\|\boldsymbol{x} - \hat{\boldsymbol{x}}\|,\ \|\boldsymbol{x}' - \hat{\boldsymbol{x}}\|\}$.

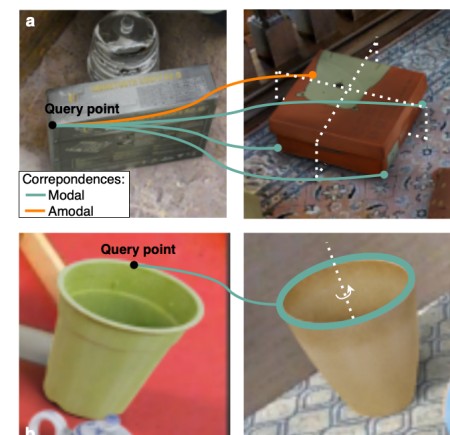

Figure 3: **Illustration of symmetries.** Row (a) shows reflective symmetry and row (b) shows rotational symmetry (white indicates the symmetry axis). Orange lines denote amodal correspondences, emerald lines modal correspondences.

With these symmetry-aware definitions, predictions are correct if they align with any symmetric equivalent: rotationally symmetric points are judged by distance to their orbit, and mirror-symmetric points by distance to the closer counterpart. This yields a fair metric that respects object symmetries and enables robust evaluation of correspondence.

Table 1: Comparison of dataset statistics across existing benchmarks. In contrast to prior datasets, HouseCorr3D provides large-scale, realistic imagery with dense 3D semantic keypoint (*i.e.,* sem. kp) annotations across 50 object categories, enabling both correspondence learning and pose estimation. $\sim$ indicates that only azimuth annotations are provided, discretized into bins of 30 degrees.

| Dataset | main goal | #images | #instances | #classes | sem. kp | realistic | pose |
|---------|-----------|---------|------------|----------|---------|-----------|------|
| NOCS-CAMERA25 | 6D pose | 300k | 1085 | 6 | ✗ | ✗ | ✓ |
| NOCS-REAL275 | 6D pose | 8k | 42 | 6 | ✗ | ✓ | ✓ |
| OmniNOCS | 6D pose | 380k | 97 | 20 | ✗ | ✓ | ✓ |
| Omni6DPose | 6D pose | 807k | 4743 | 149 | ✗ | ✓ | ✓ |
| Spair71k | sem. corr. | 71k | N/A | 18 | ✓ (2D) | ✓ | $\sim$ |
| HouseCorr3D | sem. corr. | 277k | 280 | 50 | ✓ (3D) | ✓ | ✓ |

**Dataset statistics.** HouseCorr3D contains 277k images across 50 object categories and 280 unique instances, extracted from the photo-realistic synthetic scenes of Omni6DPose (Lin et al., 2023). For each class, we annotate between 2 and 10 semantic 3D keypoints directly on the mesh, which can be projected consistently into any rendered view and used to construct correspondence pairs across instances or scenes. Thanks to the realism of the underlying Omni6DPose scenes, our dataset includes clutter, natural lighting, and partial occlusions, making it visually close to real data. As summarized in Table 1, HouseCorr3D is the first large-scale benchmark to combine realistic imagery, 3D semantic keypoints, and support for both correspondence learning and pose estimation, thereby filling the gap between correspondence-focused datasets such as SPair71k (which only provides sparse 2D keypoints and coarse azimuth annotations in 30° bins) and pose-focused datasets such as Omni6DPose. More detailed statistics about the dataset can be found in Appendix A.1.

## 4 METHOD

Our goal is to move beyond NOCS-style pose recovery toward *category-level 3D correspondences*: predicting, from a single image, the canonical 3D location of each pixel in a way that is semantically aligned across all instances of a category. To achieve this, we introduce **CaNOCS (Canonicalized NOCS)**, a framework that combines explicit pose estimation with a learned morphable prior. Unlike NOCS, which applies a fixed normalization to each instance, CaNOCS learns a deformable canonical space that ensures semantic consistency across objects of the same category.

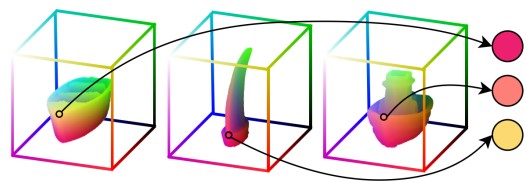

Figure 4: **NOCS lacks semantic consistency**: The same semantic point on three boat instances is highlighted, yet their NOCS color coding differs, showing inconsistency.

### 4.1 REPRESENTATION OF THE MORPHABLE 3D PRIOR

A central component of CaNOCS is the *3D morphable prior*, a category-level canonical shape space. It consists of a canonical mesh capturing the common structure of a category, along with a learned deformation model that adapts it to individual instances. We call it a *prior* because all predictions are constrained to be deformations of this canonical representation, ensuring semantic consistency across instances.

**Canonical Shape Representation.** Traditional mesh-only representations are often fragile and difficult to optimize directly, typically requiring manual interventions such as remeshing (Goel et al., 2022; Yang et al., 2021). To overcome this limitation, we employ a *hybrid volumetric–mesh design*. This integrates the strengths of implicit and explicit 3D models. Concretely, following Sommer et al. (2025) the category-level shape is represented as a signed distance field $\phi_{sdf}$, which provides the flexibility to model intricate geometries. Through Differentiable Marching Tetrahedra (DMTet) (Shen et al., 2021), the SDF is efficiently transformed into a watertight mesh in a differentiable manner by evaluating SDF values on a tetrahedral grid. We denote the mesh as $M = \{V, E\}$, where $V = \{v_i \in \mathbb{R}^3\}_{i=1}^{|V|}$ are the vertices and $E = \{(v_i, v_j)_e\}_{e=1}^{|E|}$ the edges. This formulation enables the

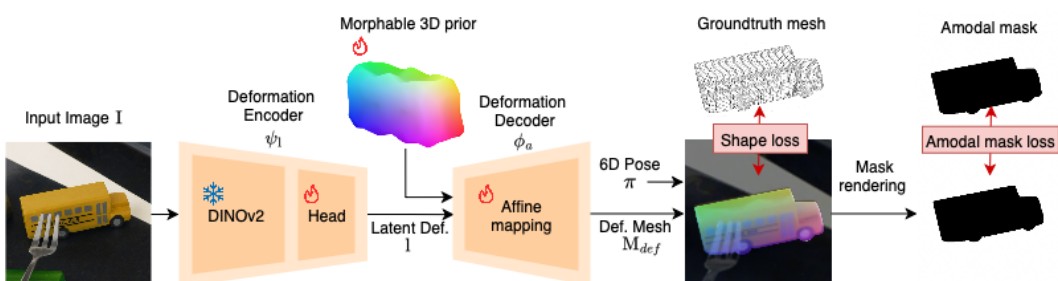

Figure 5: **Pipeline Overview.** Given an RGB image and an estimated 6D pose, CaNOCS predicts a dense 2D–3D correspondence map by reconstructing the object in a canonicalized category space. The model takes as input a cropped object image and outputs a canonical mesh with semantically aligned correspondences, plus per-pixel canonical coordinates. The key innovation over prior work is the use of a *morphable prior*—a deformable representation shared across a category—that guarantees that the same canonical vertex corresponds to the same semantic part across different object instances.

use of mesh-based priors and regularizations, such as enforcing rigidity constraints during deformation learning.

**Instance-Specific Deformations.** To adapt this canonical mesh to specific instances, we apply an affine deformation field. Unlike Zheng et al. (2021), where deformations are applied directly to the signed distance field, we act on the template mesh vertices. This avoids repeatedly extracting meshes for each instance and is thus more computationally efficient. Formally, we define a mapping $\phi_a(\boldsymbol{v}, \mathrm{l}) : \mathbb{R}^3 \times \mathrm{L} \to \mathbb{R}^3$, which displaces each vertex $\boldsymbol{v}$ of the canonical mesh according to the instance-specific latent code $\mathrm{l}$:

$$\phi_a(\boldsymbol{v}, \mathrm{l}) = \alpha(\boldsymbol{v}, \mathrm{l}) \odot \boldsymbol{v} + \delta(\boldsymbol{v}, \mathrm{l}), \quad \alpha, \delta \in \mathbb{R}^3. \tag{1}$$

Here, $\alpha$ and $\delta$ are produced by an MLP that takes both the vertex coordinate $\boldsymbol{v}$ and the latent code $\mathrm{l}$ as input. The latent code $\mathrm{l}$ itself is computed from the input image $\mathrm{I}$ by a deformation encoder $\psi_\mathrm{l}$ built from a DINOv2 backbone with a light convolutional head. This code parameterizes vertex-wise displacements, enabling the mesh to morph into the observed instance while preserving semantic alignment across the category. The resulting instance-adapted mesh is written as $\mathrm{M}_{def}(\mathrm{I}) = \{\mathrm{V}_{def}(\mathrm{I}), \mathrm{E}\}$, where each deformed vertex is given by $\mathrm{V}_{def}(\mathrm{I}) = \{\phi_a(\mathrm{v}_i, \psi_\mathrm{l}(\mathrm{I}))\}_{i=1}^{|\mathrm{V}|}$. In this way, CaNOCS differs fundamentally from NOCS: instead of a rigid normalization, it learns a canonicalization that is flexible yet semantically consistent.

### 4.2 TRAINING OBJECTIVES

CaNOCS is trained on object-centric images with supervision from existing pose datasets (*e.g.,* Omni6DPose), while HouseCorr3D correspondences are reserved for evaluation. We jointly optimize the template and deformation model using geometric objectives: 2D mask-based reconstruction, 3D mesh-based reconstruction, and mesh regularization. These reconstruction terms enforce consistency between rendered projections of the canonical mesh and the segmentation masks. Mesh regularization promotes rigidity in instance-specific deformations and hence maintains plausible canonicalization. Together, these objectives encourage the model to reconstruct category-consistent canonical meshes while preserving instance-specific details.

**2D Reconstruction.** We first supervise using amodal object masks. Given the predicted mask $\tilde{\mathrm{m}}(\mathrm{M}_{def}, \mathrm{I}, \pi)$ rendered from the deformed mesh $\mathrm{M}_{def}$ under pose $\pi$, we compare against the ground-truth amodal mask $\mathrm{m}$ with a pixel-wise mean squared error:

$$\mathcal{L}_\mathrm{m}(\mathrm{M}_{def}, \mathrm{I}, \pi, \mathrm{m}) = \left\| \tilde{\mathrm{m}}(\mathrm{M}_{def}, \mathrm{I}, \pi) - \mathrm{m} \right\|^2. \tag{2}$$

Additionally, we encourage overlap with the distance transform $\mathrm{m}_{\mathrm{dt}}$ of the ground-truth amodal mask:

$$\mathcal{L}_{\mathrm{mdt}}(\mathrm{M}_{def}, \mathrm{I}, \pi, \mathrm{m}_{\mathrm{dt}}) = -\tilde{\mathrm{m}}(\mathrm{M}_{def}, \mathrm{I}, \pi)\mathrm{m}_{\mathrm{dt}}, \tag{3}$$

with $m_{dt}$ encoding the distance of each pixel inside the mask to the silhouette boundary, while pixels outside the mask are zero, which prevents disconnected parts from emerging when fitted across diverse instances.

**3D Reconstruction.** To ensure accurate instance reconstructions, we use a Chamfer distance between the deformed mesh vertices $V_{def}$ and the ground-truth mesh vertices $V_{gt}$.

$$\mathcal{L}_{CD}(I, M_{def}, M_{gt}) = \frac{1}{|V_{def}| + |V_{gt}|} \left( \sum_{\boldsymbol{v}_i \in V_{def}} \|\boldsymbol{v}_i - \boldsymbol{v}'_{\chi(\boldsymbol{v}_i)}\| + \sum_{\boldsymbol{v}'_i \in V_{gt}} \|\boldsymbol{v}'_i - \boldsymbol{v}_{\chi(\boldsymbol{v}'_i)}\| \right), \quad (4)$$

where $\chi$ denotes the nearest neighbor operator.

**Template and Deformation Regularization.** Following Sommer et al. (2025), we enforce the SDF property with the Eikonal loss $\mathcal{L}_{sdf}$ (Gropp et al., 2020), penalize large deformation with an $\ell_2$ term: $\mathcal{L}_{def}$, and encourage smoothness with an edge-based regularizer loss $\mathcal{L}_{smooth}$ (Zheng et al., 2021). More details about these losses are provided in Appendix A.3.

**Full Training Loss.** Our optimization proceeds in two stages. First, we refine the category-level template using only geometric terms:

$$\mathcal{L}_{\text{geo}} = \lambda_{CD}\mathcal{L}_{CD} + \lambda_{\text{m}}\mathcal{L}_{\text{m}} + \lambda_{\text{mdt}}\mathcal{L}_{\text{mdt}} + \lambda_{sdf}\mathcal{L}_{sdf}. \quad (5)$$

Subsequently, we learn the instance deformations with the extended regularized loss:

$$\mathcal{L}_{\text{geo-reg}} = \mathcal{L}_{\text{geo}} + \lambda_{def}\mathcal{L}_{def} + \lambda_{smooth}\mathcal{L}_{smooth}. \quad (6)$$

While morphable priors capture fine-grained correspondences, we rely on a pretrained diffusion network (Lin et al., 2023) for pose estimation, as they currently provide state-of-the-art performance for 6D object pose estimation. The diffusion model provides high-level alignment of the object within the scene, while the morphable model refines the representation to establish detailed semantic correspondences. This hybrid design allows us to scale correspondence learning to realistic scene-level settings without incurring the computational overhead of running morphable models directly on entire scenes, effectively extending category-level morphable models from object-centric to scene-level correspondence learning. Key innovations of our approach include the integration of occlusion handling during training, support for non-centered bounding-box crops, and the joint use of diffusion models for global pose estimation (Lin et al., 2023) with morphable priors for fine-grained correspondence.

### 4.3 ESTIMATING CATEGORY-LEVEL 3D CORRESPONDENCES

CaNOCS is evaluated on *category-level 3D correspondence*. Each pixel is rasterized (cf. Appendix A.4) to a surface point on the predicted deformed mesh $M_{def} = \phi_a(M, l)$, obtained from the template $M$ and latent deformation $l = \psi_l(I)$, and aligned to canonical space using the pose from Lin et al. (2023).

For 3D correspondence, a ground-truth canonical point $\mathbf{x}^\star$ is matched to the nearest predicted canonical point $\hat{\mathbf{x}} \in \hat{\mathcal{X}}$:

$$\hat{\mathbf{x}}_{\chi(\mathbf{x}^\star)} = \arg\min_{\hat{\mathbf{x}} \in \hat{\mathcal{X}}} \|\mathbf{x}^\star - \hat{\mathbf{x}}\|_2, \text{ with error } d(\mathbf{x}^\star) = \|\mathbf{x}^\star - \hat{\mathbf{x}}_{\chi(\mathbf{x}^\star)}\|_2. \quad (7)$$

For 2D correspondence, each pixel is linked to a surface point on the deformed mesh via rasterization. Then, similarly to 3D, we can compute correspondence by matching nearest neighbors in the canonical space. Further discussion of correspondence evaluation, including the distinction between modal and amodal settings, is provided in Appendix A.5. Ground-truth correspondences come from HouseCorr3D, which provides both modal (visible) and amodal (including occluded) annotations. Errors are normalized by the object's bounding box and used to compute PCK.

## 5 EXPERIMENTS

We evaluate CaNOCS on the proposed **HouseCorr3D** benchmark, focusing on its ability to recover *category-level 3D correspondences*. We compare CaNOCS to baselines derived from NOCS

and recent 2D feature-matching approaches such as DINOv2, and evaluate performance using the symmetry-aware metrics introduced in Section 3. Our experiments cover quantitative results on 3D and 2D correspondences, qualitative analysis of semantic consistency across challenging categories, and ablations on the pose prediction. Additionally, we discuss failure cases in Appendix A.6, which typically arise from strong deformations relative to the template, which the model cannot capture reliably.

## 5.1 EXPERIMENTAL DETAILS

We use a pretrained ViT-S DINOv2 (Oquab et al., 2023) as our backbone. From an input resolution of $256 \times 256$, the backbone maps to a 16x16 feature map. The deformation encoder is implemented as a ResNet head that aggregates multi-scale feature maps with bottleneck blocks to produce refined latent deformation l. The deformation decoder is a coordinate-conditioned MLP head that fuses 3D point embeddings with latent deformation to predict deformations. To learn the initial template shape, we train each category-specific morphable model using the Adam optimizer (Kingma & Ba, 2015) with a learning rate of $1 \times 10^{-4}$ and a batch size of 30. Training proceeds in two stages: (i) 20 epochs optimizing the loss $\mathcal{L}_{\text{geo}}$ (Equation (5)), and (ii) 10 further epochs optimizing the extended loss $\mathcal{L}_{\text{geo-reg}}$ (Equation (6)), which includes deformation regularizers. Training on a single NVIDIA GeForce RTX 2080 takes approximately 7 hours.

## 5.2 CORRESPONDENCE METRICS

We define correspondences as mappings between observations (pixels or 3D points) and canonical reference surfaces. We distinguish three types of correspondences: **2D** (mapping from an image pixel $u \in \mathbb{R}^2$ to another pixel $v \in \mathbb{R}^2$ within the same or a paired image), **3D modal** (mapping from an observed 3D point $x \in \mathbb{R}^3$ to a canonical surface point $c \in \mathcal{C}$, restricted to visible surfaces), and **3D amodal** (mapping from any canonical surface point $c \in \mathcal{C}$ to its posed location $X \in \mathbb{R}^3$, including occluded regions). Formally, we define the correspondence function as $\phi : \mathcal{D} \to \mathcal{C}$, where $\mathcal{D}$ is the domain of observed pixels ($\mathbb{R}^2$) or 3D points ($\mathbb{R}^3$). Depending on the modality, $\phi$ may be partial (modal) or complete (amodal). We primarily evaluate with the Percentage of Correct Keypoints (PCK) metric. Ambiguities due to object symmetries (*e.g.,* left-right flips in animals or repetitive structures in man-made objects) can lead to multiple valid correspondences, which we treat separately. We provide a more detailed discussion in Appendix A.5.

## 5.3 3D CORRESPONDENCES (AMODAL + MODAL)

We first measure performance on the 3D correspondence benchmarks of HouseCorr3D, which include both *modal* (visible only) and *amodal* (including occluded) correspondences. Table 2 reports correspondence accuracy for both amodal and modal settings. Across both settings, CaNOCS consistently outperforms baselines, showing clear benefits of coupling morphable priors with semantically aligned canonical coordinates. The amodal setting is especially challenging, as it requires reasoning about occluded parts. We find that CaNOCS retains high accuracy in this regime, confirming that our supervision encourages models to predict complete, surface-level correspondences rather than only fitting to visible pixels. This further emphasizes the need for explicit 3D representations for holistic scene understanding.

Table 2: PCK@0.1 results for 3D modal, 3D amodal, and 2D correspondences on HouseCorr3D.

| Method | 🚂 | ✈ | 🚍 | 🚐 | 👕 | 📷 | 🏍 | 🔋 | 🖥 | mean |
|---|---|---|---|---|---|---|---|---|---|---|
| **3D Modal** | | | | | | | | | | |
| DINOv2+Depth | 0.071 | 0.044 | 0.065 | 0.222 | 0.122 | 0.200 | 0.134 | 0.125 | 0.132 | 0.124 |
| NOCS | 0.157 | **0.180** | 0.151 | **0.340** | 0.509 | 0.506 | 0.061 | 0.395 | 0.319 | 0.276 |
| CaNOCS | **0.241** | 0.200 | **0.172** | 0.335 | **0.749** | **0.598** | **0.052** | **0.459** | **0.327** | **0.348** |
| **3D Amodal** | | | | | | | | | | |
| CaNOCS | 0.221 | 0.262 | 0.142 | 0.315 | 0.721 | 0.307 | 0.045 | 0.412 | 0.315 | 0.305 |
| **2D** | | | | | | | | | | |
| DINOv2 | 0.104 | 0.239 | 0.307 | 0.216 | 0.154 | 0.318 | 0.196 | 0.292 | 0.247 | 0.230 |
| NOCS | 0.251 | 0.201 | 0.226 | **0.335** | 0.619 | 0.378 | 0.151 | 0.254 | **0.366** | 0.309 |
| CANOCS | **0.303** | **0.365** | **0.437** | 0.100 | **0.763** | **0.610** | **0.187** | **0.420** | 0.321 | **0.390** |

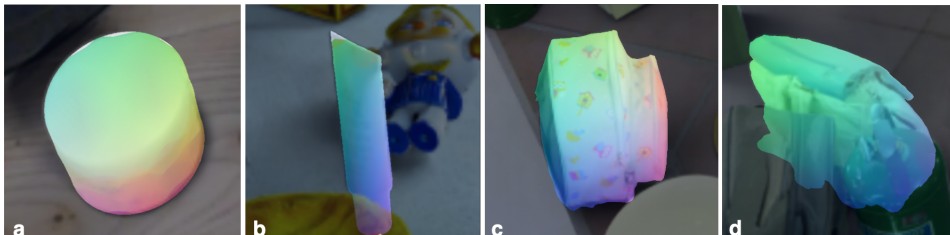

Figure 6: **Qualitative results**: (a-b) Facial cream; (c-d) Backpack. We observe that correspondences are semantically coherent across instances and views, and that amodal predictions accurately recover hidden object parts.

## 5.4 2D CORRESPONDENCES

We next evaluate in the 2D domain, where correspondences are defined between image pixels and canonical object coordinates. Table 2 compares our method with strong 2D-only baselines such as DINO-style self-supervised transformers. When ground-truth pose is available, our method leverages the underlying 3D structure to improve PCK scores, demonstrating the advantage of integrating semantic 3D information even when the evaluation takes place entirely in 2D. Notably, our approach yields significant gains on categories with large intra-class shape variation, where purely 2D methods struggle to maintain semantic alignment. Figure 6 illustrates qualitative predictions of CaNOCS.

## 5.5 ABLATIONS

We conduct ablations to disentangle the roles of pose quality and morphable priors in achieving robust 2D and 3D correspondences.

**Pose supervision:** We compare evaluation under ground-truth pose and estimated poses from Gen-Pose++ (Lin et al., 2023). Ground-truth pose improves 2D PCK, particularly on small or thin structures, showing that pose inaccuracies at test time can propagate into correspondence prediction.

**3D PCK with ground-truth pose:** We compute 3D PCK with ground-truth pose in place of Gen-Pose++ predictions. This oracle setting provides an upper bound under perfect alignment and quantifies the performance gap attributable to pose estimation noise.

Overall, these ablations confirm that accurate pose information and morphable priors are essential for robust, semantically consistent correspondences in both 2D and 3D.

Table 3: Pose ablation. Using groundtruth pose, we observe consistent better performances in terms of semantic correspondence.

| Method | 🖨 | ✈ | 🚗 | 🚌 | 👕 | 🧴 | 🏍 | 🎚 | mean |
|---|---|---|---|---|---|---|---|---|---|
| **2D PCK@0.1** | | | | | | | | | |
| CaNOCS | 0.303 | 0.089 | 0.147 | 0.110 | **0.763** | 0.610 | 0.262 | 0.420 | 0.332 |
| CaNOCS w/ GT pose | **0.321** | **0.263** | **0.255** | **0.120** | 0.638 | **0.768** | **0.359** | **0.752** | **0.436** |
| **3D Modal PCK@0.1** | | | | | | | | | |
| CaNOCS | 0.241 | 0.200 | 0.172 | 0.335 | **0.749** | 0.598 | 0.052 | 0.459 | 0.348 |
| CaNOCS w/ GT pose | **0.333** | **0.273** | **0.260** | **0.339** | 0.638 | **0.758** | **0.359** | **0.752** | **0.416** |

## 6 CONCLUSION

While prior work has advanced pose-centric canonicalization (*e.g.,* NOCS), dense but domain-specific correspondences (*e.g.,* template and deformation models), and strong 2D semantics (*e.g.,* self-supervised transformers), none has jointly achieved (i) single-image, *category-level* 3D correspondences, (ii) *semantic* consistency across large intra-class variation, and (iii) *amodal* supervision that accounts for occlusion. **CaNOCS** addresses (i) and (ii) by supervising semantically aligned canonical coordinates with morphable priors, while **HouseCorr3D** contributes (iii) at scale across 50 everyday categories. Together, they establish a new paradigm for fine-grained, correspondence-level object understanding. Looking ahead, we are excited to see how our dataset can serve as a foundation for expanding correspondence learning beyond vision benchmarks, particularly toward embodied applications in robotics where reasoning about full 3D object geometry is crucial.

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

# A APPENDIX

## A.1 ADDITIONAL DATASET STATISTICS

We rely exclusively on the realistic synthetic subset of Omni6DPose (Lin et al., 2023). Preliminary experiments showed that the real captures provide limited diversity: most categories contain at most two unique object instances, scenes are often repeated across long video sequences, and overall variation in layout is low. As a result, the number of reliable correspondences that can be established from the real subset is severely restricted.

In contrast, the synthetic pipeline offers large-scale variation in both object instances and scene composition. This diversity is crucial for learning robust 2D–3D semantic correspondences across categories. Moreover, the synthetic subset has been designed to closely mimic real-world conditions, with natural lighting, cluttered environments, and realistic occlusions. This ensures that models trained on our benchmark generalize well beyond simplified synthetic settings. Therefore, our benchmark focuses on the high-quality synthetic subset, which provides both realism and sufficient coverage for large-scale correspondence evaluation. In total, HouseCorr3D contains 277k images across 280 unique object instances from 50 categories, making it the first large-scale dataset with dense, semantically consistent 2D–3D correspondences for everyday objects.

To better illustrate the scope of annotations, Table A4 reports the number of annotated keypoints for each category, highlighting differences in semantic coverage across classes. Complementary statistics are shown in Figures A1 and A2, which respectively display the distribution of annotated keypoints per class and the number of annotated object instances. Together, these results provide a comprehensive view of the dataset's scale and diversity, supporting its suitability as a benchmark for category-level 3D correspondence.

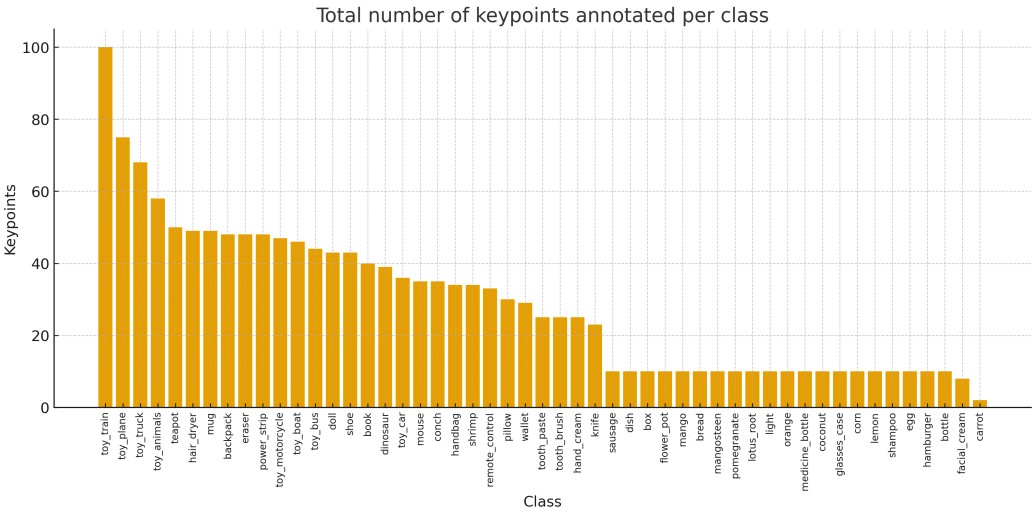

Figure A1: Total number of annotated keypoints per class.

## A.2 MESH ANNOTATION PROCESS

For mesh annotation, we convert each CAD mesh into a point cloud to facilitate visual inspection and interaction. Annotators are then provided with up to 10 3D keypoints per category that must be placed consistently across all instances. These keypoints are chosen to be semantically meaningful and geometrically well-defined: rather than marking the center of a continuous surface, annotators focus on distinctive structures such as corners, edges, wheel centers, handles, or wing tips. This strategy ensures that annotated points are both discriminative and reliably transferable across different instances of a category.

To guarantee annotation quality, each instance was independently annotated by two annotators. If the discrepancy between their placements exceeded 5% of the object's bounding-box diagonal, the

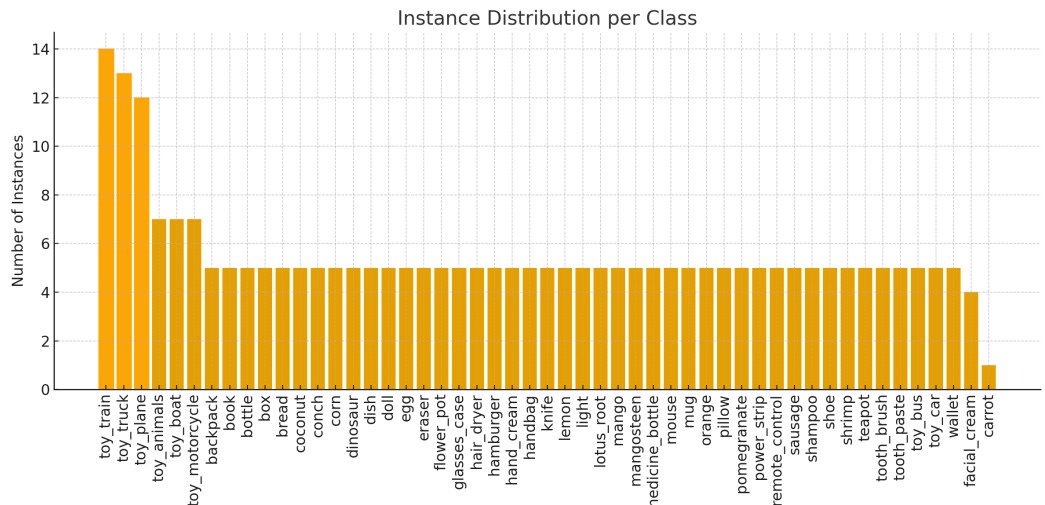

Figure A2: Total number of annotated instances per class.

annotators were asked to re-annotate until consensus was achieved. This iterative procedure reduced noise and led to high-quality annotations consistent across the dataset.

In addition, the reference mesh for each category was annotated first, and subsequent instances were aligned to this reference using a 3D interface. This alignment step further reduced ambiguities and ensured that annotations across different instances adhered to the same semantic standard. Overall, this process yields a compact yet semantically robust set of 3D keypoints that serve as the foundation for our correspondence benchmark.

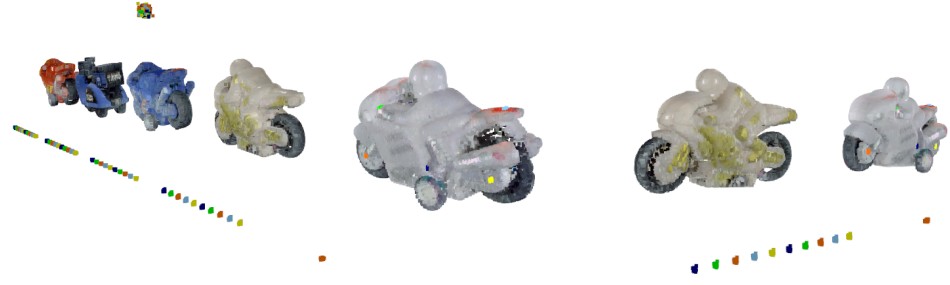

(a) **Overview of the annotation tool.** With all the points that need to be assigned.

(b) Close up view of the instance requiring annotation (left) and the reference instance (right).

Figure A3: **Annotation process illustration.** Using an interactive 3D interface, 5 instances are aligned, and 3D keypoints have to be assigned to the respective meshes, based on the annotation on the reference instance (here on the right).

## A.3   ADDITIONAL LOSSES

Learning accurate correspondences requires not only supervision on visible matches but also strong geometric regularization to stabilize training and enforce plausible shapes. To this end, we use additional loss terms that constrain the learned deformation and shape representation.

**Eikonal loss.** To enforce the signed distance function (SDF) property, we adopt the Eikonal regularizer (Gropp et al., 2020), which encourages unit-norm gradients of the implicit function. Because gradients are only reliable near the extracted surface, we additionally sample auxiliary points $\mathcal{P}_{sdf}$

throughout the canonical space:

$$\mathcal{L}_{sdf}(\mathrm{M}, x) = \left(\|\nabla\phi_{sdf}(x)\|_2 - 1\right)^2, \quad x \in \mathcal{P}_{sdf}. \tag{A1}$$

This prevents degenerate fields and stabilizes the geometry across unseen regions.

**Deformation regularizer.** To avoid arbitrary or excessive deformations, we penalize $\ell_2$ deviations of vertices from the category template:

$$\mathcal{L}_{def}(\mathrm{M}, \mathrm{M}_{def}, \mathrm{I}) = \frac{1}{|\mathrm{V}|} \sum_{\boldsymbol{v} \in \mathrm{V}} \left\|\boldsymbol{v} - \phi_a(\boldsymbol{v}, \psi_1(\mathrm{I}))\right\|^2. \tag{A2}$$

This term encourages learned shapes to remain close to the canonical prototype while still allowing instance-specific variation.

**Smoothness regularizer.** Finally, we promote locally coherent deformations by enforcing smooth displacements across neighboring vertices, following Zheng et al. (2021):

$$\mathcal{L}_{smooth}(\mathrm{M}, \mathrm{M}_{def}, \mathrm{I}) = \frac{1}{|\mathrm{E}|} \sum_{\boldsymbol{i},\boldsymbol{j} \in \mathrm{E}} \frac{\left\|[\boldsymbol{i} - \phi_a(\boldsymbol{i}, \psi_1(\mathrm{I}))] - [\boldsymbol{j} - \phi_a(\boldsymbol{j}, \psi_1(\mathrm{I}))]\right\|_2}{\|\boldsymbol{i} - \boldsymbol{j}\|_2}. \tag{A3}$$

This regularizer suppresses spurious local distortions while still allowing non-rigid articulation.

Together, these terms ensure that the learned representation respects the SDF property, stays anchored to a canonical template, and maintains smooth, realistic deformations.

### A.4 MASK RENDERING

In computer graphics, *rendering* denotes the entire pipeline of generating an image from a 3D scene, while *rasterization* is the specific step that converts geometric primitives into pixels on the image plane. Since our rendering pipeline consists solely of rasterization, in this paper we use the two terms interchangeably. Concretely, we project the deformed mesh $M^{\mathrm{def}}$ into the camera view using the ground-truth pose $\pi$. Each triangle of the mesh is then discretized to the pixel grid, and pixels falling inside a triangle are marked as foreground. Visibility is resolved using a $z$-buffer, so only the closest surface along each camera ray contributes to the mask. In this way, each pixel corresponds to a unique surface point on the deformed mesh. This rasterization step provides the binary object masks used in our 2D reconstruction losses and also forms the link between image pixels and 3D surface points for correspondence evaluation. Because the pipeline only involves rasterization (without additional shading, lighting, or texturing), the procedure is both efficient and fully differentiable, which is essential for training CaNOCS.

### A.5 DISCUSSION ABOUT CORRESPONDENCE EVALUATION

**Modal vs. Amodal masks** We distinguish between *modal* and *amodal* correspondences in 3D. Modal correspondences are defined only on the subset of surface points that are visible from a given viewpoint, mapping observed 2D pixels to their canonical surface counterparts. In contrast, amodal correspondences extend this mapping to the full object surface, including parts that may be occluded or self-hidden. Modal evaluation reflects how well a method can align observed geometry with a canonical template and is directly comparable to tasks such as 2D keypoint transfer. Amodal evaluation goes further: it measures whether a model has learned a complete category-level shape prior that can predict correspondences even for unobserved surfaces. This distinction is critical for downstream tasks that require holistic understanding, such as shape completion, scene reasoning, or part-level manipulation. In 2D, we are restricted to image pixels, which by definition correspond only to visible regions; there is no ground-truth notion of a pixel for an occluded surface. In 3D, however, we can explicitly represent the canonical surface $\mathcal{C}$ and predict both visible and occluded points across poses. This makes it possible to evaluate amodal correspondences, providing a stronger test of a model's ability to infer complete, semantically consistent shapes across instances.

**Symmetries** In the main paper Section 3, we defined symmetry-aware correspondence errors for rotational and reflective cases. While the theoretical formulation includes both discrete $C_N$ groups

and continuous rotations ($N \rightarrow \infty$), in practice, we approximate continuous rotational symmetry by discretizing the angle into $N = 16$ bins. This corresponds to the cyclic group $C_{16}$, which balances computational tractability with angular resolution ($22.5°$). Empirically, increasing the discretization (*e.g.,* $N = 32$) did not yield measurable gains given the annotation precision of HouseCorr3D.

During evaluation, the correspondence error for a predicted point is computed against all symmetric equivalents, and the minimum is retained, ensuring predictions aligned with any symmetric variant are correct. The discretization further enables efficient vectorization across batches, avoiding costly minimization over continuous angles. During evaluation, the correspondence error for a predicted point is computed against all symmetric equivalents, and the minimum is retained. This ensures that predictions aligned with any symmetric variant are considered correct. The discretization also enables efficient vectorization across batches, avoiding costly minimization over continuous angles. This approximation is not a practical limitation since (i) the chosen resolution is finer than the annotation granularity, (ii) rotationally symmetric categories are faithfully captured, and (iii) reflective symmetries are handled exactly by considering mirrored counterparts. A natural extension would be to handle *continuous* rotational symmetry exactly. One option is to optimize directly over the rotation angle (e.g., gradient search or closed-form projection). Another equivalent formulation would be to compute the distance from a predicted point to the rotation plane, which geometrically corresponds to the continuous orbit of the keypoint. Such approaches could tighten error bounds for highly symmetric categories and provide a unified treatment of exact and approximate symmetries.

**2D correspondences.** In the 2D setting, both query and target keypoints are ideally visible in their respective images. However, due to occlusion and viewpoint changes, this assumption often fails, making nearest-neighbor search in 2D unreliable. As illustrated in Figure 1, our 3D morphable prior enables reasoning about *amodal 2D correspondences*, *i.e.,* correspondences that remain valid even when points are not directly visible.

The procedure is as follows:

   (i) Each pixel is linked to a surface point on the deformed mesh via rasterization.

  (ii) The query point is mapped to the canonical space through the deformable prior.

 (iii) It is then transferred to the target deformed mesh and projected back into the target image.

This results in a 2D–3D–2D correspondence pipeline, which is only possible thanks to the shared canonical representation that ties all views and instances together.

**Additional results** In addition to the results reported in the main paper (Sections 5.3 and 5.4), we provide in Table A1 the complete set of quantitative results for our method, covering all categories of HouseCorr3D. These extended results complement the main text by offering a more fine-grained view of per-class performance. Importantly, we observe the same overall trends as in the main paper. This consistency arises because the categories highlighted in the main figures were chosen at random, rather than being selected to favor particular outcomes. Thus, the additional results confirm that our observations hold uniformly across the entire benchmark and are not biased by the choice of examples shown in the main paper.

**Metric** In the main paper, we report results using the Percentage of Correct Keypoints (PCK) metric. For completeness, we provide a formal definition below:

$$\text{PCK@}\alpha = \frac{1}{N} \sum_{i=1}^{N} \left( \|\hat{\mathbf{x}}_i - \mathbf{x}_i\|_2 < \alpha \cdot \text{diag}_{\text{bbox}} \right) \tag{A4}$$

where $N$ is the number of keypoints, $\alpha$ the threshold, and $\text{diag}_{\text{bbox}}$ the bounding-box diagonal. We report both 2D PCK and 3D PCK (modal and amodal).

Table A1: Per-category correspondence accuracy (PCK@0.1) for 2D, 3D modal, and 3D amodal settings.

| | backpack | book | bottle | box | bread | coconut | dinosaur | dish | doll | egg | eraser | facial_cream | glasses_case | hair_dryer | hamburger | hand_cream | handbag | knife |
|---|---|---|---|---|---|---|---|---|---|---|---|---|---|---|---|---|---|---|
| **2D** | | | | | | | | | | | | | | | | | | |
| DINOv2 | 0.104 | 0.214 | 0.558 | 0.158 | 0.154 | 0.499 | 0.318 | 0.222 | 0.140 | 0.132 | 0.268 | 0.318 | 0.262 | 0.296 | 0.239 | 0.343 | 0.193 | 0.213 |
| CaNOCS | 0.303 | 0.325 | 0.701 | 0.700 | 0.763 | 0.575 | 0.089 | 0.147 | 0.110 | 0.141 | 0.377 | 0.610 | 0.342 | 0.262 | 0.791 | 0.423 | 0.230 | 0.307 |
| **3D Modal** | | | | | | | | | | | | | | | | | | |
| DINOv2+depth | 0.071 | 0.156 | 0.267 | 0.127 | 0.122 | 0.300 | 0.102 | 0.121 | 0.082 | 0.045 | 0.125 | 0.200 | 0.192 | 0.145 | 0.160 | 0.188 | 0.107 | 0.130 |
| CaNOCS | 0.241 | 0.399 | 0.660 | 0.738 | 0.749 | 0.572 | 0.037 | 0.363 | 0.047 | 0.086 | 0.350 | 0.598 | 0.348 | 0.250 | 0.772 | 0.482 | 0.208 | 0.233 |
| **3D Amodal** | | | | | | | | | | | | | | | | | | |
| CaNOCS | 0.221 | 0.381 | 0.767 | 0.665 | 0.721 | 0.659 | 0.050 | 0.592 | 0.040 | 0.072 | 0.307 | 0.655 | 0.308 | 0.184 | 0.688 | 0.444 | 0.206 | 0.216 |

| | lemon | lotus_root | mango | mangosteen | medicine_bottle | mouse | orange | pillow | pomegranate | power_strip | remote_control | sausage | shampoo | shoe | shrimp | tooth_brush |
|---|---|---|---|---|---|---|---|---|---|---|---|---|---|---|---|---|
| **2D** | | | | | | | | | | | | | | | | |
| DINOv2 | 0.398 | 0.432 | 0.590 | 0.628 | 0.526 | 0.212 | 0.815 | 0.172 | 0.590 | 0.306 | 0.292 | 0.390 | 0.889 | 0.276 | 0.316 | 0.517 |
| CaNOCS | 0.170 | 0.352 | 0.383 | 0.210 | 0.755 | 0.485 | 0.518 | 0.494 | 0.315 | 0.487 | 0.420 | 0.251 | 0.821 | 0.410 | 0.216 | 0.685 |
| **3D Modal** | | | | | | | | | | | | | | | | |
| DINOv2+depth | 0.190 | 0.176 | 0.246 | 0.285 | 0.428 | 0.140 | 0.672 | 0.096 | 0.335 | 0.180 | 0.125 | 0.209 | 0.516 | 0.114 | 0.095 | 0.401 |
| CaNOCS | 0.123 | 0.337 | 0.263 | 0.110 | 0.814 | 0.469 | 0.400 | 0.504 | 0.113 | 0.439 | 0.459 | 0.280 | 0.801 | 0.371 | 0.122 | 0.662 |
| **3D Amodal** | | | | | | | | | | | | | | | | |
| CaNOCS | 0.055 | 0.237 | 0.134 | 0.263 | 0.775 | 0.349 | 0.498 | 0.286 | 0.263 | 0.376 | 0.412 | 0.241 | 0.843 | 0.373 | 0.143 | 0.600 |

| | tooth_paste | toy_boat | toy_bus | toy_car | toy_motorcycle | toy_plane | toy_train | wallet |
|---|---|---|---|---|---|---|---|---|
| **2D** | | | | | | | | |
| DINOv2 | 0.425 | 0.247 | 0.307 | 0.216 | 0.140 | 0.196 | 0.239 | 0.247 |
| CaNOCS | 0.415 | 0.244 | 0.437 | 0.100 | 0.279 | 0.187 | 0.365 | 0.321 |
| **3D Modal** | | | | | | | | |
| DINOv2+depth | 0.326 | 0.065 | 0.222 | 0.134 | 0.070 | 0.044 | 0.132 | 0.146 |
| CaNOCS | 0.458 | 0.172 | 0.335 | 0.052 | 0.190 | 0.200 | 0.327 | 0.329 |
| **3D Amodal** | | | | | | | | |
| CaNOCS | 0.428 | 0.142 | 0.315 | 0.045 | 0.212 | 0.262 | 0.315 | 0.303 |

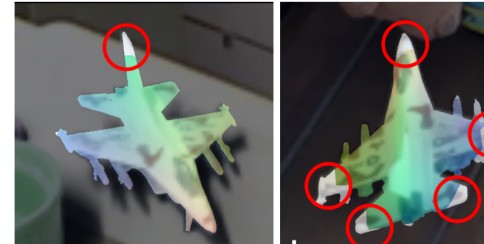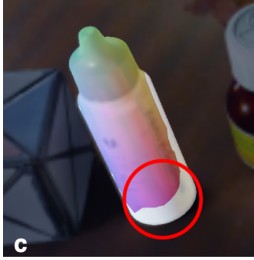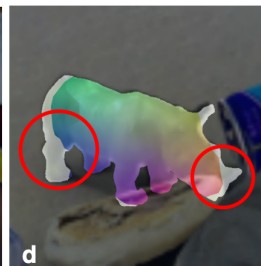

Figure A4: **Failure cases.** Instances requiring large deformations or containing thin structures often lead to errors, as the deformation decoder cannot extend sufficiently to capture the true geometry. Typical artifacts include under-deformed extremities, truncated parts, or floating correspondences, highlighted by red circles.

## A.6 FAILURE CASES

Despite the overall robustness of our method, several limitations can be observed in challenging scenarios. A first source of error arises from inaccurate pose estimation from Lin et al. (2023). Since canonical alignment is a prerequisite for predicting consistent correspondences, pose misalignments can propagate through the pipeline and lead to incorrect predictions. A second limitation concerns the deformation decoder. The learned deformations are constrained by both the template representation and the distribution of training data. As a result, objects that exhibit high intra-class variability, or that contain fine-scale structures not well captured in the template, often cannot be deformed adequately. This is especially evident for thin or elongated extremities such as airplane wings, bottle tips, or animal legs, where the predicted deformation either underestimates the required displacement or in some rare cases collapses the geometry entirely, as highlighted in Figure A4. Finally, the model may fail in cases where very large non-linear deformations are required. Since the decoder is trained to interpolate within the observed shape distribution, extrapolations to unseen structural variations remain difficult. Consequently, regions that extend far beyond the canonical template tend to remain under-deformed, leading to visible artifacts such as truncated parts or floating geometry. While these errors are relatively rare, they underscore the inherent trade-off between enforcing a shared canonical prior and maintaining sufficient flexibility to capture extreme shape variations across object instances.

## REPRODUCIBILITY STATEMENT

To ensure full reproducibility of our work, we will release all code and data used in this paper. The complete processing pipeline, including scripts for dataset preparation and annotation generation, will be made publicly available on GitHub. Our training and inference code for the proposed model will be provided in the same repository, along with configuration files and instructions for reproducing all experiments reported in the paper. The dataset itself, including the annotated 3D meshes and projected 2D keypoints, will be released on Hugging Face for easy access and long-term hosting. In addition, we will provide helper functions to compute the 3D correspondence metrics introduced in this paper, ensuring that results can be evaluated in a consistent and standardized manner.

## LLMs USAGE

We make limited use of large language models (LLMs) to support the writing process of this paper. In particular, LLMs were employed for three purposes only:

(i) rephrasing sentences to improve readability or conciseness,

(ii) shortening paragraphs that were overly lengthy, and

(iii) organizing preliminary notes or fragmented ideas into coherent paragraphs.

All technical content, methods, experiments, and analyses remain entirely our own.

ETHICS STATEMENT

All authors of this paper have read and adhere to the ICLR Code of Ethics.[1] The dataset annotations were carried out entirely by the authors of this work. No external annotators or paid workers were involved. All annotation efforts were voluntary and conducted by researchers with domain expertise, ensuring both high quality and ethical treatment of annotators.

The dataset consists of synthetic renderings of CAD models and does not involve personal data, identifiable human subjects, or sensitive information. As such, it does not raise concerns regarding privacy, security, or legal compliance. Our methodology does not produce harmful content, and our benchmark is intended purely for academic research in computer vision and related fields.

We further note that the dataset and code will be released under a research-friendly license, ensuring transparency and reproducibility while discouraging misuse. To the best of our knowledge, this work raises no ethical issues regarding discrimination, fairness, or conflicts of interest.

---

[1] https://iclr.cc/public/CodeOfEthics

Table A4: Number of annotated keypoints per object class.

| Class | Number of keypoints |
| --- | --- |
| backpack | 10 |
| book | 8 |
| bottle | 2 |
| box | 2 |
| bread | 2 |
| carrot | 2 |
| coconut | 2 |
| conch | 7 |
| corn | 2 |
| dinosaur | 8 |
| dish | 2 |
| doll | 10 |
| egg | 2 |
| eraser | 10 |
| facial_cream | 2 |
| flower_pot | 2 |
| glasses_case | 2 |
| hair_dryer | 10 |
| hamburger | 2 |
| hand_cream | 5 |
| handbag | 10 |
| knife | 5 |
| lemon | 2 |
| light | 2 |
| lotus_root | 2 |
| mango | 2 |
| mangosteen | 2 |
| medicine_bottle | 2 |
| mouse | 7 |
| mug | 10 |
| orange | 2 |
| pillow | 6 |
| pomegranate | 2 |
| power_strip | 10 |
| remote_control | 7 |
| sausage | 2 |
| shampoo | 2 |
| shoe | 9 |
| shrimp | 7 |
| teapot | 10 |
| tooth_brush | 5 |
| tooth_paste | 5 |
| toy_animals | 10 |
| toy_boat | 8 |
| toy_bus | 10 |
| toy_car | 8 |
| toy_motorcycle | 10 |
| toy_plane | 9 |
| toy_train | 10 |
| toy_truck | 10 |
| wallet | 7 |

