# OpenReview forum: "CaNOCS: Category-Level 3D Correspondence from a single image"
_ICLR.cc/2026/Conference — ICLR 2026 Conference Withdrawn Submission_

### Official Review · Reviewer_aCoL · 2025-10-18

**Soundness:** 3
**Presentation:** 2
**Contribution:** 2
**Rating:** 4
**Confidence:** 2

**Summary:**

This work proposes to achieve 3D correspondence by predicting from images the 3D location of each pixel, such that this location is semantically aligned across all instances of that category. It is done by introducing a new dataset HouseCorr3D, derived from the existing Omni6DPose. The key innovation in the model the authors suggest is including a morphable shape priors that are shared across given categories.

**Strengths:**

1. The problem and main idea was clearly stated and described.

2. Although there was no supervision on the learnable shape priors, the authors suggest/show in Section 5.3 that it aids in matching occluded regions, by learning the canonical representation of each category.

3. Most of the time, the model outperforms the baselines.

**Weaknesses:**

Main issues:

1. The main drawback for me is that important evaluations are missing. Although DenseMatcher (ICLR 2025 Spotlight) was mentioned in line 159, as another 3D correspondence work - the authors only claim that: "but they still fall short of providing image-to-canonical or
amodal correspondences across diverse categories.". This claim requires experimental justification. In addition, regarding the "diverse categories" part, the suggested model CaNOCS is only able to provide correspondence for the categories it was trained on. So I'm not sure it is fair to claim that DenseMatcher falls short with regards to category diversity.

2. There a a lot of loss terms (7!) suggested in the paper, and it it is not clear which are necessary and which are not. This challenges the understanding the model and the real objective of the training, and is especially problematic for future work.
 An ablation study investigating and justifying each loss term will be helpful.

3. Currently the model was trained and tested on the same data distribution (data from Omni6DPose and new suggested annotations). First, since the annotations are generated and are not fully manual - a manual inspection/verification of a random portion of dataset will help insuring the GT annotations are valid. Also, lines 309-310 state that the model was trained on Omni6DPose and tested on HouseCorr3D. Since HouseCorr3D derived from Omni6DPose, please clarify how the data was split and that there is minimal data pollution. Make sure there are not the same images/instances. Lastly, I wonder how well this model generalizes to new instances from outside of the data distribution. Please see the Questions section below.

Minor:
3. It would be helpful if the amodal/modal distinction be defined earlier in the text.
4. Figure 4: circle highlighs are wrong. Figure 5 is in low resolution. Please consider inluding this figure as a vector graphic (e.g. pdf/svg).
5. Notation in Equation 1: alpha and delta are functions or numbers in R^3?
6. M_def is defined differently in lines 360 and 303 (?).
7. 2D correspondence is mentioned under the title "ESTIMATING CATEGORY-LEVEL 3D CORRESPONDENCES" in Section 4.3.
8. Line 402: it is not clear how 3D corespondence is measured. You define a match given a GT canonical point in Equation 7. How do you measure PCK here given a predicted point? Please clarify.

**Questions:**

A further analysis of the key innovation will clarify more about its contribution. I wonder how the model depends on the learned morphable shape priors. For example, how will the model perform on images from the same category but from image samples from outside the dataset distribution?

And what about different categories? A model trained on the category 'bus', will produce reasonable results on car images?

---

### Official Review · Reviewer_Atkk · 2025-10-27

**Soundness:** 3
**Presentation:** 3
**Contribution:** 2
**Rating:** 4
**Confidence:** 5

**Summary:**

This paper discusses a limitation of the original NOCS regarding the semantic consistency. The authors argue that NOCS lacks semantic consistency across instances within the same object category. They handle this problem by introducing a new method that learns category-level morphable shape priors. Given an RGB image, a deformed mesh is predicted based on the image and the shape prior. It enables multiple types of correspondence estimation, including 3D correspondences and 2D correspondences. A new dataset, HouseCorr3D, is introduced to evaluate the method, which has dense semantic 2D–3D correspondences across 50 household object categories.

**Strengths:**

The paper raises a valid point of NOCS about semantic understanding and reframes the task as category-level 3D correspondence. While not revolutionary, it is a reasonable extension of existing work on category-level object pose estimation towards more semantically consistent 3D representations.

The proposed HouseCorr3D benchmark offers accurate object poses and consistent semantic keypoints across object instances. The object symmetry is also considered, which is important in object pose estimation.

The CaNOCS framework combines established techniques and learns a deformed mesh based on an input RGB image and a morphable 3D shape prior. The experimental results show that the method achieves better results compared with NOCS and DINOv2.

**Weaknesses:**

The idea of semantic 3D correspondences is not entirely new, and the contributions regarding this are a bit overclaimed. The problem of semantic correspondences has been investigated in various settings [1, 2, 3, 4, 5]. Some datasets, such as Keypointnet and the one in Densematcher, also offer annotations of 3D semantic correspondences. The general idea is quite similar to that proposed in [5], which also learns a point-level deformation from a shape prior.

The argument of “NOCS lacks semantic consistency” is confusing and not well-explained. In Fig.4, the authors show that the NOCS colors are different for the three different instances, which thus shows a semantic inconsistency problem. However, this may not be the case. The shapes of different instances within the same object category are different in the normalized object coordinate system, and the different RGB visualizations are thereby reasonable. This doesn’t mean NOCS itself is semantically inconsistent.  An argument such as “previous datasets in category-level object pose estimation lack annotations of 3D semantic correspondences” would be more precise.

The experimental results are not convincing enough. Only NOCS and DINOv2 are compared in PCK@0.1. None of the follow-up works of NOCS are evaluated. It is unclear if the proposed category-level morphable shape priors can facilitate category-level object pose estimation without the relevant experiments.

[1] Zhu, Junzhe, et al. "Densematcher: Learning 3d semantic correspondence for category-level manipulation from a single demo." arXiv preprint arXiv:2412.05268 (2024).

[2] You, Yang, et al. "Keypointnet: A large-scale 3d keypoint dataset aggregated from numerous human annotations." *Proceedings of the IEEE/CVF Conference on Computer Vision and Pattern Recognition*. 2020.

[3] Lou, Yujing, et al. "Human correspondence consensus for 3d object semantic understanding." *European Conference on Computer Vision*. Cham: Springer International Publishing, 2020.

[4] Wandel, Krispin, and Hesheng Wang. "SemAlign3D: Semantic Correspondence between RGB-Images through Aligning 3D Object-Class Representations." Proceedings of the Computer Vision and Pattern Recognition Conference. 2025.

[5] Tian, Meng, Marcelo H. Ang Jr, and Gim Hee Lee. "Shape prior deformation for categorical 6d object pose and size estimation." *European Conference on Computer Vision*. Cham: Springer International Publishing, 2020.

**Questions:**

Please refer to my concerns in Weaknesses

---

### Official Review · Reviewer_v46S · 2025-10-29

**Soundness:** 2
**Presentation:** 4
**Contribution:** 2
**Rating:** 2
**Confidence:** 4

**Summary:**

The paper tackles the problem of predicting dense 3D correspondences between an input RGB image and a canonical 3D space, aiming for semantic consistency across different object instances within the same category. Unlike prior approaches focused on 6D pose estimation or instance-level correspondence, this work formulates the task at the category level, enabling richer scene understanding. The authors propose CaNOCS, a framework that leverages morphable shape priors to guide per-pixel predictions of canonical 3D coordinates, ensuring alignment across instances. To support this task, they introduce HouseCorr3D, a new benchmark dataset with dense 2D–3D correspondences across 50 object categories, complete with CAD models and amodal masks. Extensive experiments show that their method outperforms baselines such as NOCS and DINOv2-based models, demonstrating improved semantic consistency and generalization.

**Strengths:**

- The paper introduces a novel (though not the first one, see weakness) problem setting: category-level 3D dense correspondence prediction from a single RGB image, which extends beyond traditional 6D pose estimation.

- The authors contribute a new benchmark dataset, HouseCorr3D, with dense 2D–3D correspondences across 50 object categories, which will likely benefit the community.

- Experimental results demonstrate clear performance improvements over NOCS and other baselines, validating both the approach and the dataset.

**Weaknesses:**

- First, many of the citations in the paper appear to have been generated by a large language model, resulting in inaccuracies. For example, in line 179, OmniNOCS is incorrectly cited as *Lin et al.* when it should be attributed to *Krishnan et al.* Similarly, Omni6DPose should be cited as: *Zhang, Jiyao, et al. "Omni6DPose: A Benchmark and Model for Universal 6D Object Pose Estimation and Tracking." European Conference on Computer Vision. Cham: Springer Nature Switzerland, 2024.* This use of LLM-generated references contradicts the claim made in line 996, raising concerns about the soundness and reliability of the paper.

- Second, the paper omits discussion of several relevant works on category-level correspondences, such as KeypointNet [1], KeypointDeformer [2], and CSM [3], which already address and handle category-level correspondences. This omission weakens the novelty and contextual grounding of the paper’s contributions.

- Third, the core idea of using a template mesh is not novel and has been explored in prior works, including Mesh R-CNN [4] and NeuralCage [5]. The dataset itself may not reach the acceptance bar of ICLR.

References:

[1] KeypointNet: A Large-scale 3D Keypoint Dataset Aggregated from Numerous Human Annotations

[2] KeypointDeformer: Unsupervised 3D Keypoint Discovery for Shape Control

[3] Canonical Surface Mapping via Geometric Cycle Consistency

[4] Mesh R-CNN

[5] Neural Cages for Detail-Preserving 3D Deformations

**Questions:**

See weakness.

---

### Official Review · Reviewer_y4in · 2025-11-01

**Soundness:** 3
**Presentation:** 4
**Contribution:** 3
**Rating:** 4
**Confidence:** 4

**Summary:**

The paper introduces HouseCorr3D, a new benchmark of dense, amodal 2D–3D semantic correspondences across 50 everyday object categories, with symmetry-aware evaluation. It further proposes CaNOCS, which augments NOCS with a category-level morphable 3D prior to enforce cross-instance semantic consistency from a single image. CaNOCS uses diffusion-based pose prediction for global alignment and the morphable prior for fine-grained correspondence, trained with silhouette, Chamfer, and regularization losses. On HouseCorr3D, CaNOCS notably outperforms NOCS- and DINOv2-based baselines for 3D (modal & amodal) and 2D correspondence (PCK@0.1), and ablations show sensitivity to pose accuracy. Overall, the dataset + method aim to shift from pose-centric canonicalization to category-level correspondence

**Strengths:**

1. Hybrid SDF to mesh canonical template with image-conditioned affine deformations yields semantically consistent canonicalization across instances; pragmatic two-stage training with geometric losses is technically sound.
2. This paper releases the HouseCorr3D benchmark with amodal labels and symmetry-aware metrics across 50 categories—filling a measurable gap left by NOCS datasets.

**Weaknesses:**

1. CaNOCS is mainly based on the Common3d [1]. The experiments should include the comparison between Common3d [1] and CaNOCS.
2. Previous work [2] had studied the category-level correspondence through a  2D-3D-2D cycle and used category-level 3D semantic corresponded keypoints. The pipeline for finding the category-level correspondences needs more discussion.
3. The experiments use the proposed HouseCorr3D benchmark to validate the performance over the performance, which is limited. The experiments could also be conducted on previous correspondence benchmarks like SPair-71k and PF-Pascal.

[1] Sommer L, Dünkel O, Theobalt C, et al. Common3d: Self-supervised learning of 3d morphable models for common objects in neural feature space[C]//Proceedings of the Computer Vision and Pattern Recognition Conference. 2025: 6468-6479.

[2] You Y, Li C, Lou Y, et al. Understanding Pixel-Level 2D Image Semantics With 3D Keypoint Knowledge Engine[J]. IEEE Transactions on Pattern Analysis and Machine Intelligence, 2021, 44(9): 5780-5795.

**Questions:**

See weaknesses.

---

### Note · Authors · 2025-11-12

I have read and agree with the venue's withdrawal policy on behalf of myself and my co-authors.